# PIVQGAN: Posture and Identity Disentangled Image-to-Image Translation via Vector Quantization

## Abstract

One popular objective for the image-to-image translation task is to independently control the coarse-level object arrangements (posture) and the fine-grained level styling (identity) of the generated image from two exemplar sources. To approach this objective, we propose PIVQGAN with two novel techniques in the framework of StyleGAN2. First, we propose a Vector-Quantized Spatial Normalization (VQSN) module for the generator for better pose-identity disentanglement. The VQSN module automatically learns to encode the shaping and composition information from the commonly shared objects inside the training-set images. Second, we design a joint-training scheme with self-supervision methods for the GAN-Inversion encoder and the generator. Specifically, we let the encoder and generator reconstruct images from two differently augmented variants of the original ones, one defining the pose and the other for identity. The VQSN module facilitates a more delicate separation of posture and identity, while the training scheme ensures the VQSN module learns the pose-related representations. Comprehensive experiments conducted on various datasets show better synthesis image quality and disentangling scores of our model. Moreover, we present model applications beyond posture-identity disentangling, thanks to the latent-space reducing feature of the leveraged VQSN module.

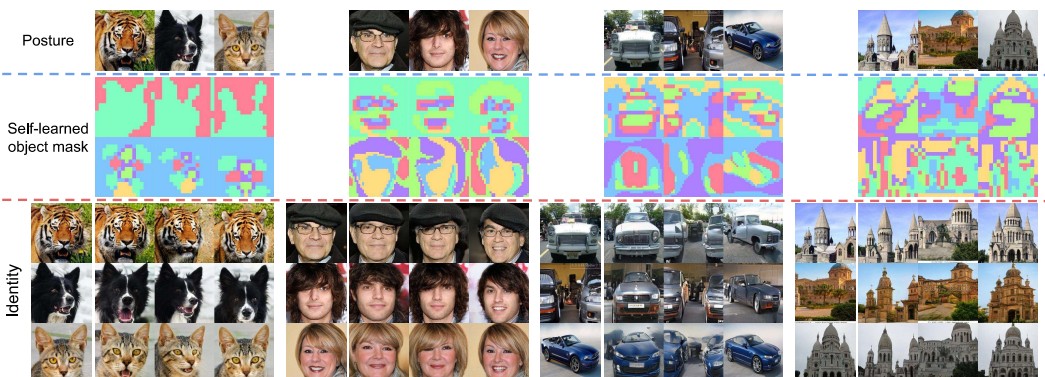

Figure 1: Unsupervised image-to-image translation results of PIVQGAN with disentangled posture and identity control. In each panel, the first row has input pose images, and the first column has referential identity images. The second and third rows are "segmentation-like" masks automatically learned by PIVQGAN, and bottom-right are the synthesized images.

## 1 Introduction

We study a sub-task of image-to-image translation, which we call posture-and-identity disentangled translation. We view an image as a combination of multiple objects with varied scales, then define *posture* as the shape and composition of the larger-scale objects, and *identity* as both the color/texture

of the objects and the shape of the finer-scale objects. Given two input images, one defines posture, and the other defines identity. Our goal is to develop a model that transfers the first image's stance to the second one's identity. This task has excellent potential in real-life applications such as illustration synthesis and motion driving for content-creation jobs. Note that a close task has been widely studied as content-style disentangled image synthesis, e.g. Choi et al. (2020); Kwon & Ye (2021); Lee et al. (2021); Liu et al. (2021; 2020a); Kotovenko et al. (2019); Kazemi et al. (2019). We find the terminology of "content-style" can be ambiguous sometimes, as both posture and identity has content and style related properties. In contrast, "posture-identity" are relatively precise to the task and datasets that we are working on, including human, animal, and animation faces.

Training an image-to-image translation model usually requires image-level or set-level supervision, as represented by StarGAN-v2: Isola et al. (2017); Wang et al. (2018); Park et al. (2019); Zhu et al. (2017); Li et al. (2018); Lee et al. (2018); Liu et al. (2019); Choi et al. (2020). However, it is expensive to obtain paired data and extra labeling information in real-world scenarios. More recently, fully unsupervised methods gain more research interests such as CLUIT: Baek et al. (2021); Kwon & Ye (2021); Lee et al. (2021), where no labeling information but only a set of images are needed. However, these methods lack controllability on disentangled image attributes, and lack robustness on the degree of disentanglement among datasets. For our task of pose-identity translation, the limitations from prior works motivate us to seek a fully unsupervised method that is also controllable and robust. It is hard to train a model with precise control over posture and identity when there are no guidance labels of an image such as a segmentation or skeleton map, which are used by Nirkin et al. (2018); Lee et al. (2020); Kocabas et al. (2020); Jin et al. (2020). Nonetheless, the need for such a segmentation label inspires us to develop a module that can learn a segmentation-like representation from the unlabeled image data in a self-supervised manner, therefore boosting the translation performance, as shown in Fig. 1.

Our key idea is to let the image generator automatically learn a semantic mask of the posture image. Different regions in the mask represent a semantic object in the image. The synthesis is then based on the objects' shape and composition in the mask. To facilitate the idea, we devise a Vector-Quantized Spatial Normalization layer (VQSN) that combines vector-quantization (Oord et al., 2017) and spatially-adaptive normalization (Park et al., 2019). Specifically, we first leverage the VQ mechanism to reduce a feature-map into a certain number of object embeddings. Each embedding corresponds to one common object shared among images of the same domain (e.g. eye, mouth, and nose for face dataset; wall, window, and roof for building dataset). We then get a spatial object mask that we refer to when performing an affine transformation on the feature-map. Each spatial location is transformed differently according to its corresponding object.

We show that, with a carefully designed self-supervision training scheme, VQSN can automatically learn meaningful object embeddings that provide useful posture information. The resulting model, which we call PIVQGAN, enables a precise control on posture and identity of the image generation, and outperforms existing baselines in terms of visual quality and translation accuracy. Besides, we show a robust performance of PIVQGAN even on unseen out-of-domain images, where we further demonstrate its promising application beyond pose-identity translation, including zero-shot image style-transfer and semantic-aware image-editing.

## 2 RELATED WORK

### 2.1 UNSUPERVISED IMAGE TRANSLATION

Well established image-to-image translation models mostly require labeled data to be trained, e.g. Isola et al. (2017); Zhu et al. (2017); Liu et al. (2017); Kim et al. (2017); Huang et al. (2018); Kim et al. (2019); Liu et al. (2019); Choi et al. (2020). Instance level paired images are required by models with an autoencoder training scheme like employed by Isola et al. (2017); Park et al. (2019); Liu et al. (2021), and coarse domain labels are needed for uni- or multi-modal translation models trained via cycle-consistency scheme as in Zhu et al. (2017); Huang et al. (2018); Choi et al. (2018).

However, the need for labels often becomes a bottleneck in real-world applications, limiting both the model performance and the application scenarios. Therefore, fully unsupervised image translation has drawn more research attention recently. One kind of unsupervised methods acquire pseudo-labels by image clustering (Baek et al. 2021, Bahng et al. 2020), another kind directly generates

paired images via style-transfer from a small portion of labeled data (Liu et al., 2021). These methods lack robustness, as unintended translation results may occur when clustering algorithms fail to produce consistent clusters, or the synthesized pair data has low quality.

Focused on posture-and-identity image translation, methods without label-supervision are developed based on a reliable prior provided by StyleGAN (Karras et al. 2019). Kwon & Ye (2021) proposed a diagonal attention structure which aims to encode the object shaping information. Lee et al. (2021) developed a contrastive self-supervised training scheme, where a model is trained to implicitly learn and transfer visual features without explicit domain separation. However, the disentangled latent features from these methods are not well interpretable and controllable, thus lacks practicality in real world scenarios. Our work combines the strength from both perspectives. We propose a structural design that retains a more explicit object shaping representation, and leverages self-supervision to enforce the learning without labeling data.

## 2.2 DISENTANGLED IMAGE SYNTHESIS

A broader task of disentangled image synthesis usually aims to learn one vector representation for the images in an image domain, where each vector entry controls one mutually independent visual attribute. Autoencoder-based methods, e.g. Higgins et al. (2017); Burgess et al. (2018); Kim & Mnih (2018); Esmaeili et al. (2018), approach disentanglement via the total-correlation objective, while GAN based models, e.e. Chen et al. (2016); Lin et al. (2020); Liu et al. (2020b), rely on maximizing a mutual-information loss. However, the vector representation has a limited capacity, and the per-attribute disentanglement is still hard to achieve. In this paper, we study the coarse-level disentanglement between two sets of attributes, which can be seen as the very first step in a hierarchically-disentangled task as proposed by Esmaeili et al. (2018); Singh et al. (2019).

The coarse-level disentanglement is usually approached by the inductive biases of the layer designs and model structures. For example, Karras et al. (2019); Karnewar & Wang (2020); Liu et al. (2020c) showed that controlling the attributes of a generated image from multiple scales in the generator is effective. Park et al. (2019) developed SPADE which applies different affine transformations to different spatial regions of the feature-map, to highlight different object shapes and diversify the object appearances. Kwon & Ye (2021) proposed to learn "content" branch beside the original "style" branch in StyleGAN, and uses a one-channel spatial mask to guide the content information. Kim et al. (2021) further replaced the vector representation in the "style" branch with spatial masks to embed better shaping information to the generated images.

We also bet on the structural bias to gain better control over the posture generation. However, unlike previous methods, our model learns the posture information in an interpretable, transferable, and editable way while disentangled well from the identity attributes.

## 3 METHODS

In this section, we first describe PIVQGAN's synthesis flow. Then we introduce Vector-Quantized Spatial Normalization (VQSN). Finally, we talk about the self-supervised training scheme and all the objective functions.

### 3.1 MODEL OVERVIEW

PIVQGAN consists of three models, the discriminator $D$ and the generator $G$ (modified with VQSN layers) from StyleGAN2, and an encoder $E$ that converts images to vectors in $G$'s latent space.

In each training iteration, $D$ and $G$ are first trained as GAN (Goodfellow et al. 2014), then $G$ and $E$ are trained for GAN inversion tasks (as in Xia et al. 2021) like an Autoencoder. A similar joint-training schema without dedicated self-supervision is also explored by Kim et al. (2021) in StyleMapGAN. Such a joint-training is required in our case to implement the proposed self-supervisions as described in Sec 3.3.

We separate the generation of posture and identity attributes on different convolutional layers of $G$. Specifically, given that $G$ synthesizes an image at the resolution of $256 \times 256$, a base feature-map of size $4 \times 4$ is up-scaled six times by 14 conv-layers, each conv-layer modulated by a corresponding

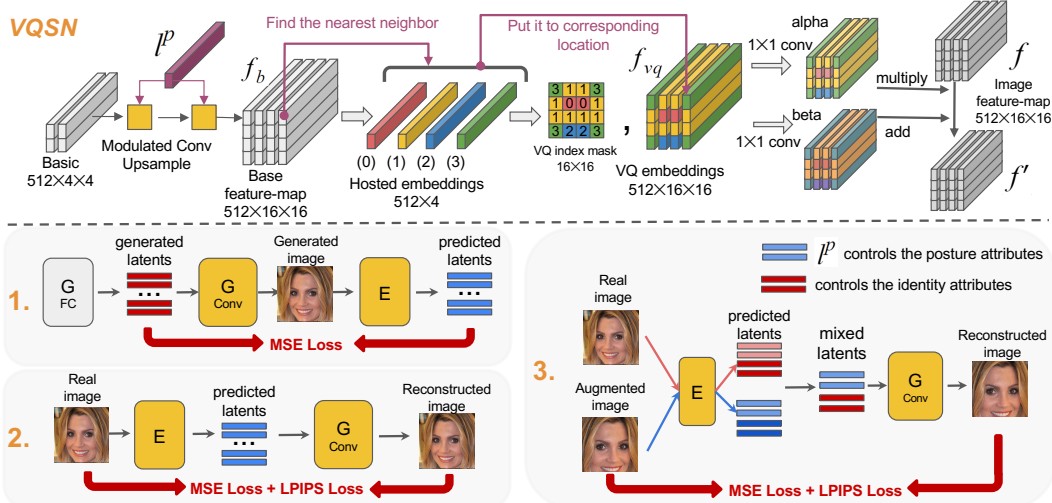

Figure 2: **Top**: Illustration of the VQSN module. (1) A base feature-map is quantized by a certain number (e.g. 2 or 6) of trainable embeddings. (2) The VQ embeddings are used to perform the spatial-wise affine transformation on image feature-maps. **Bottom**: Training scheme of PIVQGAN. Apart from the regular GAN training (omitted), we train an encoder together with the convolution part of the generator with self-supervision tasks. In (3), we do latents-grafted reconstruction to help the model disentangle pose/identity attributes at different layers.

latent vector. We take 2 to 3 layers between 2nd to 7th layers (varied on datasets) as the *pose-layer*. The *pose-layer*s are forced to learn only posture-related information, which has to be independent of the other layers. Meanwhile, given an image, $E$ is trained to generate the 14 latent vectors that can let $G$ reconstruct the image. The goal of this setting is to let the latent vectors for pose-layers encode only the pose information, while the rest carry other attributes.

In inference time, $E$ gets two latent sets $l^1_{[1:14]}, l^2_{[1:14]}$ from two images $i_1, i_2$. And we graft the latent sets by swapping the *pose-layer*s' vectors to transfer the posture or identity features between the two images. For example, suppose layer 4 to 6 are *pose-layer*s, the grafted latent sets $[l^1_{[1:3]}, l^2_{[4:6]}, l^1_{[7:14]}]$ will let $G$ generate a image with the identity features of $i_1$, while having the posture of $i_2$. Moreover, we can also randomly sample $l$ from the latent space of $G$, for more diverse translations.

## 3.2 VECTOR-QUANTIZED SPATIAL NORMALIZATION

With the hope of letting the selected *pose-layer*s learn only pose-related features while not influencing the rest layers to maintain a high synthesis quality, we propose the VQSN layer. VQSN is a plug-in module that works on the output feature-map of common convolutional layers.

The VQSN module takes two inputs: a vector $l^p \in \mathbb{R}^{512}$ (from StyleGAN's latent space) that defines the posture attributes, and an image feature-map $f \in \mathbb{R}^{c \times h \times w}$, e.g. $c, h, w = (512, 16, 16)$. And it outputs a revised feature-map $f'$ in the same shape. There are two parts inside the VQSN module. First, a *pose generator* has a trainable basis in the shape of $512 \times 4 \times 4$ and three modulated convolutional layers (like a tiny StyleGAN2 generator). This pose generator uses $l^p$ to up-sample the basis into a base feature-map $f_b \in \mathbb{R}^{c \times h \times w}$. Second, as shown in Fig. 2-left, $f_b$ is quantized into $f_{vq}$ according to a certain number of trainable embedding vectors. Finally, $f_{vq}$ is converted into *alpha* and *beta* via $1 \times 1$ convolutional layers, and performs affine transformation on $f$ to get the output $f'$.

Two intuitions guide our design of VQSN. First, we hope to compress the representation capability of the pose-layers so that it has no room to carry unwanted shape-invariant information. Vector-quantization fits well in this case, for it reduces the continuous representation space into a few discrete embeddings. The VQ module categorizes each spatial location into one object. And the categorization process naturally defines the shape of each object and the composition of all the objects.

Second, to highlight shape-variant information, we want to treat each spatial location differently in a feature-map (e.g. the eye area has a value distribution that is different from the mouth area). We apply unique affine-transformation at different pixel locations according to the quantized object mask to address such a difference.

### 3.3 SELF-SUPERVISED TRAINING SCHEME

The encoder $E$ aims at projecting an image into the latent space of $G$, where $G$ can faithfully reconstruct the input image. There are two main training stages when we jointly train $E$ and $G$. First, we sample latent vectors $l_{gt} \in \mathbb{R}^{b \times 14 \times 512}$ from the mapping network (Karras et al. 2019) in $G$ with random noise as input. Here we train the decoder network in $G$ and $E$ to reconstruct the sampled latent vectors:

$$\mathcal{L}_{\text{latent}} = \mathbb{E}[||E(G(l_{gt})) - l_{gt}||_2] \tag{1}$$

$\mathcal{L}_{\text{latent}}$ ensures $E$ and $G$ to be well communicated, thus helps $E$ to always produce in-domain latent vectors in $G$'s latent space. Second, we sample real images $i_{real}$ and train $E$ and $G$ to reconstruct them with LPIPS loss (Zhang et al. 2018):

$$\mathcal{L}_{\text{rec}} = \mathbb{E}[||i_{real} - G(E(i_{real}))||_2] + LPIPS(i_{real}, G(E(i_{real}))) \tag{2}$$

Importantly, while the VQSN module captures large-scale posture features well (e.g. head direction), it can hardly capture smaller-scale pose information, such as mouth and eyes openness, by itself. To help VQSN capture those detailed postures, we apply a series of augmentations to the real images and train $E$ and $G$ on the augmented image reconstruction task. As shown in Fig. 4, we use shape-oriented augmentations, including random cutout, flipping, and re-scaling, to create pose and identity image pairs $[i_{org}, i_{aug}]$. Then, as shown in Fig. 2-bottom-(3), we train $E$ and $G$ on reconstruction tasks from the grafted latents (again suppose layer 4 to 6 are pose layers):

$$l_{[1:14]}^{org} = E(i_{org}); \quad l_{[1:14]}^{aug} = E(i_{aug})$$
$$\mathcal{L}_{\text{aug}} = LPIPS(G([l_{[1:3]}^{org}, l_{[4:6]}^{aug}, l_{[7:14]}^{org}]), i_{aug}) + LPIPS(G([l_{[1:3]}^{aug}, l_{[4:6]}^{org}, l_{[7:14]}^{aug}]), i_{org}) \tag{3}$$

$\mathcal{L}_{\text{aug}}$ makes sure that only the selected pose-layers encode these augmentation-related shape information while the rest do not carry them. With this training scheme, VQSN automatically learns the proper semantics of the commonly shared objects between training images.

## 4 EXPERIMENTS

In this section, we describe the evaluation metrics and experiment results. We first compare PIVQ-GAN to the state-of-the-art methods, then present additional analysis and demonstrate applications of it for further insights. In each experiment, we train all the models at $256 \times 256$ resolution. Our model is trained five times in each setting, and the median score is reported.

**Baselines.** We compare PIVQGAN with both latest and some earlier baseline models that perform examplar-based image-to-image translation. Including DATGAN by Kwon & Ye (2021), CLUIT by Lee et al. (2021), SNI by Alharbi & Wonka (2020), StarGAN-v2 by Choi et al. (2020), MUNIT by Huang et al. (2018), and TUNIT by Baek et al. (2021). For models that are not open-source, we use the scores reported from the authors.

**Datasets.** We show qualitative results of PIVQGAN on five datasets covering a wide range of image domains, including CelebA Liu et al. (2015), AFHQ Choi et al. (2020), LSUN car and church Yu et al. (2016), and Anime-Girls Li et al. (2021). And we conduct quantitative comparisons with other models on CelebA and AFHQ datasets. We do not use any labeling data; only images are needed to train our model.

**Metrics.** For quantitative comparisons, we use Frechét inception distance (FID, Heusel et al. 2017) and learned perceptual image patch similarity (LPIPS, Zhang et al. 2018) to evaluate the synthesized image quality and diversity. Perceptual path length (PPL, Karras et al. 2019) is used to measure the smoothness and linearity of the latent space of $G$. We also include a simple new metric called *mix-back LPIPS*. Given two images $i_a, i_b$, we exchange their pose by the models to get $i_{ab}, i_{ba}$ ($i_{ab}$ has the identity of $i_a$ and pose of $i_b$), then we exchange their pose again to get $i_{aba}, i_{bab}$. If pose and identity are perfectly disentangled, and the image synthesis quality is high, $i_{aba}$ should look exactly the same as $i_a$, so does $i_{bab}$ and $i_b$. For all these metrics, a lower score means better performance.

## 4.1 COMPARISON TO BASELINES

| AFHQ | FID | m-LPIPS | r-LPIPS | PPL |
|---|---|---|---|---|
| MUNIT | 68.32 | 0.842 | 0.512 | - |
| TUNIT | 48.92 | 0.721 | 0.491 | - |
| SNI | 16.02 | 0.481 | 0.385 | 65.22 |
| StarGAN-v2 | 19.93 | 0.671 | 0.436 | 97.83 |
| DATGAN | 16.09 | 0.472 | 0.378 | 63.44 |
| ours | **12.54** | **0.463** | **0.373** | **58.63** |
| **CelebA** | | | | |
| MUNIT | 85.74 | 0.856 | 0.537 | - |
| TUNIT | 36.88 | 0.766 | 0.451 | - |
| SNI | 17.45 | 0.461 | 0.382 | 57.63 |
| StarGAN-v2 | 22.35 | 0.671 | 0.405 | 72.26 |
| DATGAN | 18.11 | 0.472 | 0.407 | **48.12** |
| ours | **12.65** | **0.447** | **0.331** | 51.32 |

Table 1: Comparison to baselines.

| AFHQ | FID | m-LPIPS | r-LPIPS |
|---|---|---|---|
| VQ-2 | 16.18 | 0.493 | 0.401 |
| VQ-3 | **10.98** | 0.512 | **0.358** |
| w/o augment | 12.32 | 0.532 | 0.366 |
| w/o VQ | 12.93 | 0.518 | 0.362 |
| VQ-1 | 12.54 | **0.463** | 0.373 |
| **CelebA** | | | |
| VQ-2 | 12.65 | **0.447** | **0.331** |
| VQ-3 | **10.82** | 0.451 | 0.341 |
| w/o augment | 12.72 | 0.522 | 0.346 |
| w/o VQ | 13.53 | 0.538 | 0.352 |
| VQ-1 | 12.46 | 0.449 | 0.339 |

Table 2: Ablation study.

StarGAN-v2 and CLUIT have the best visual quality among the baseline models, so we only show quality comparison with these two models on the AFHQ dataset in Fig. 3. We consider AFHQ data harder than CelebA because it has more prominent identity feature variance with different animal breeds. PIVQGAN outperforms its competitors in both translation accuracy and synthesis quality. Our model produces a sharper image with more precise details and correct coloring in overall image quality. In terms of translation accuracy, PIVQGAN overcomes some common defects shared by prior methods. Specifically, models before CLUIT can hardly maintain a good identity consistency. StarGAN-v2, TUNIT, and MUNIT even generate animals with the wrong species. CLUIT fails to translate the detailed posture attributes, including eye and mouth open degree and ear fold degree. In contrast, PIVQGAN not only translates these postures well but also captures the fine-grained identity features better, such as the texture of leopard, the ear color of fox, and the eye color of the cats, where other models fail to capture.

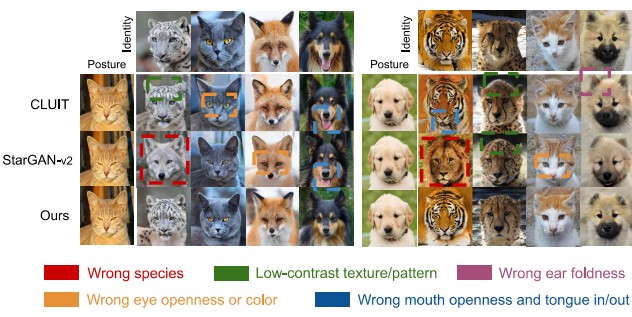

Figure 3: Qualitative comparison of PIVQGAN and other baselines on AFHQ dataset.

To compute FID, we use images from the testing set as pose and identity reference images, randomly matching them to get 10000 input pairs and generate 10000 testing samples. These samples are then used to compute FID with all the training set images. The same input pairs are also used to calculate the mix-back LPIPS. We also use LPIPS to measure the models' reconstruction performance, where we use the same testing image as both pose and identity reference, and compute the mean LPIPS of 10000 samples. To calculate PPL, we randomly sample two images from the testing set as a pair and get latent vectors from them, then interpolate the two latent sets to compute the score. We also use 10000 sample pairs and report their mean value.

Table. 1 shows the metrics comparison results. For both datasets, our model stands out with almost 25% improvement on FID over the previous state-of-the-arts, indicating a better image quality and a more diverse coverage on identity and posture features. The better mix-back LPIPS shows that our model separates and preserves the pose and identity features well.

## 4.2 ANALYSIS OF MODEL COMPONENTS

**VQSN vs Gumbel-softmax.** To learn a segmentation-like representation that captures the object shape and composition feature, Gumbel-softmax (GS, Jang et al.) is also a viable approach besides our employed Vector-Quantization. We also implemented the GS version of our spatial transfor-

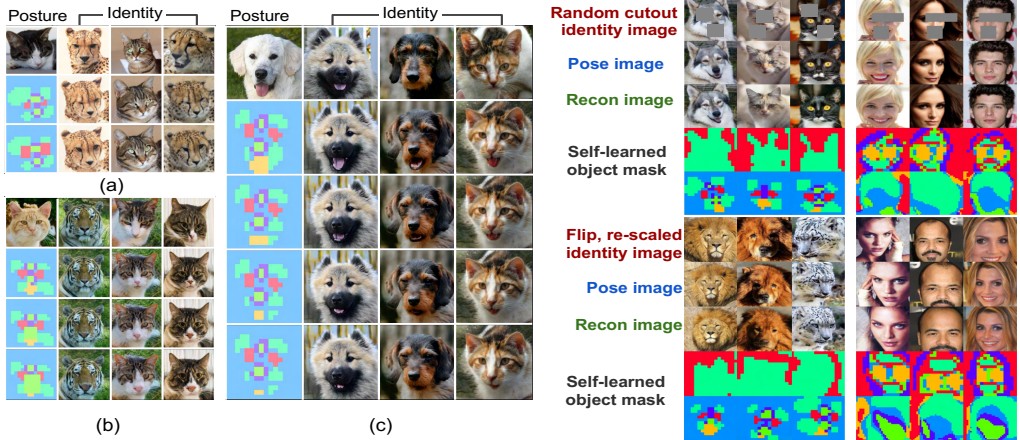

Figure 4: **Left:** Object-mask editing. **Right:** Various augmentation methods used on the model's identity inputs, enabling the model to learn the augment-variant attributes from only posture inputs.

mation module. We use the differentiable soft-max trick in GS to classify each pixel location into one of the object labels. Unfortunately, with GS, the model is unable to learn interpretable semantic masks as VQSN does. Moreover, the object mask from GS looks like random noise and leads to a worse image synthesis quality of the model. Nonetheless, we think GS is worth more study; a better parameter tuning and structure design may lead to viable performance.

**VQSN structure design.** We explore multiple structural settings of our VQSN module to have a better understanding of its performance. Among various trials, we present three notable settings:

1) VQ-1: the basic design, where the object-mask is got from a base feature-map generated from the input latent vector and modulated conv-layers.

2) VQ-2: the base feature-map is generated from the previous object mask's VQ embeddings with modulated conv-layer and the current latent vector.

3) VQ-3: the VQ-embeddings will first concatenate to the image feature-map, then generate the affine transformation parameters.

The intuition for VQ-2 is to get hierarchically coarse-to-fine object masks over multiple pose-layers. So the object-mask is generated based on the previous layer's output mask. For example, the first VQ layer learns the large-scale object information such as head region and background region; then, the second VQ layer can learn eye, mouth, and nose shapes inside the head region. For VQ-3, we aim to smooth the affine transformation on object contour areas because the object mask has a sharp difference at object boundary. Therefore, we concatenate the spatially discrete VQ embeddings with the continuous image feature-map before using the VQ embeddings to revise the image feature-map.

Interestingly, VQ-2 does not do what we expected. Instead, it leads to worse visual semantics on object masks and worse performance across the metric. We hypothesize this depends on the image visual properties of the datasets. For example, as shown in Fig. 4-right and Fig. 5-left, on AFHQ, the VQ layers first learn head shape then smaller objects. In reverse order, on CelebA, the model first learns eye, mouth, and nose objects, then captures head and hair in the next layer. We do not extend more on it in this paper but will further study the reason behind it.

Meanwhile, VQ-3 improves the image quality as expected. However, the pose translation performance slightly downgrades. The concatenated image feature-map may introduce noise to the VQ module and weaken its ability to learn object shaping features. The quantitative results on AFHQ are presented in Sec. 4.1. While VQ-3 has better FID and reconstruction LPIPS, it performs marginally worse than VQ-1 in disentangling and preserving the pose and identity features.

**Shape-related image augmentation.** Self-supervised reconstruction task with image augmentation (Fig. 2-right) is the major reason that PIVQGAN captures the small-scale object semantics at VQSN layers. PIVQGAN can hardly capture the mouth and eye pose without this training scheme, which

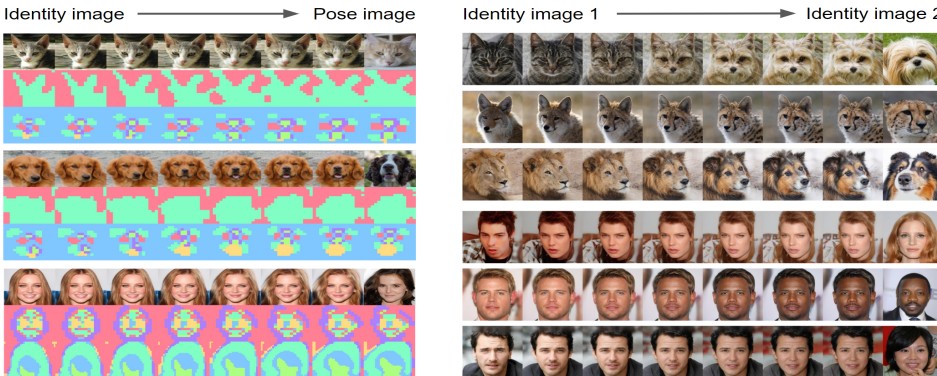

Figure 5: **Left**: VQSN is controlled by the latent vectors, thus it has a smooth posture transaction when interpolating between two pose references. **Right**: When fixing the pose-latents and interpolating only the identity-latents, PIVQGAN shows a smooth transaction on identity-related attributes.

failed by other baseline models. Both the VQ module and the augmented training bring better translation performance, as can be found according to the mix-back LPIPS score in Table. 2.

### 4.3 QUALITATIVE EVALUATION AND APPLICATIONS

**Latent space smoothness.** In the VQSN module, we still let a vector control the synthesis of the object-mask. The design is to get a smooth latent space of the object-mask. Thus we can generate continuous transactions between different postures. Fig. 5-right shows the interpolation between two posture vector sets. In each panel, the left-most image defines the identity and the initial posture, and the right-most image defines the target posture. Both inputs are real images from the testing set. We can see that both the generated images and the generated object-masks have a smooth transaction. On the third panel, we show one problem of PIVQGAN that remains to be solved. On CelebA, our model learns the hairstyle as a posture attribute rather than an identity attribute because the hairstyle is more of a shape-related property than a shape-invariant one like color and texture. Such a problem is hard to address by neither the VQSN module nor the augmentation training scheme.

Fig. 5-right further shows the smoothness of the identity latent space in PIVQGAN and also reflects the excellent disentanglement between posture and identity features of our model. The left-most image defines the pose and the initial identity in each row, while the right-most one defines the target identity. Both inputs are images from the testing set. All the synthesized images remain of high quality while maintains the original posture well. On CelebA, PIVQGAN generates expressive human faces among the interpolations. On AFHQ, the interpolated images present the breed-blending visuals between the species. It shows the potential of our model for creative material synthesis.

**Object-mask editing.** After training, the encoder $E$ combined with the VQSN layer in $G$ becomes a segmentation network. Given an image, the combined model generates object masks that highlight each component of the image. Of course, the produced masks are far less accurate than actual segmentation maps. However, these object-masks still get the essential semantics of the input image. And by editing the masks, we can create novel images from $G$ that follow our revisions.

Fig. 4-left presents some editing examples with testing images as referential inputs. In Fig. 4-left-(a), we find that the red region represents the eyes, so if we enlarge the red parts, the eyes are also getting bigger in the generated images. Similarly, the green parts correlate well with the nose in the images, so if we change the size of the green part, we achieve some funny pictures of tigers and cats having big noses. One last example is that the yellow part highlights the mouth area, so if we gradually remove the yellow part, we can see that the generated animals have their mouths closed.

**Robust on out-domain images.** We consider the robustness to unseen out-domain images to be an essential feature of PIVQGAN. The model performance is inevitably downgraded in prior works when the testing images have unseen texture, color, or objects. However, our model can maintain its high image synthesis quality as long as the out-domain images are not too far away from the training image domain. Two factors contribute to such robustness. Firstly, the joint training scheme between

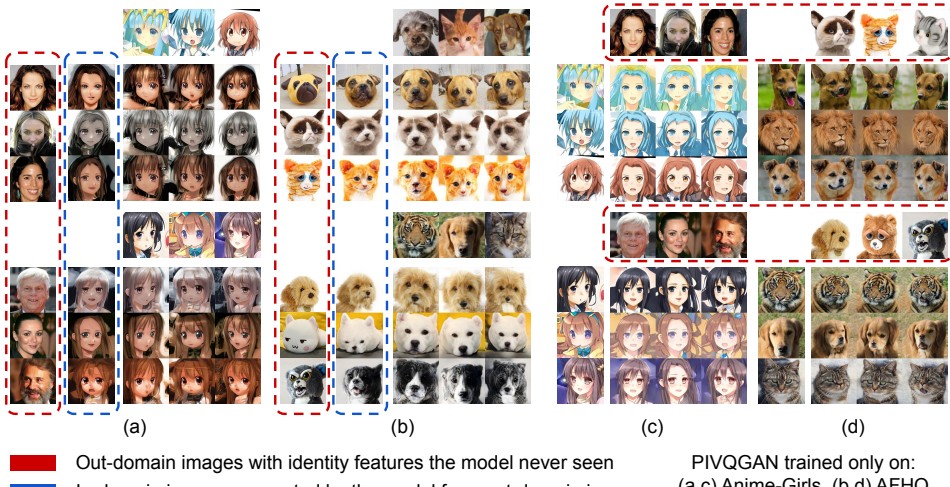

Figure 6: PIVQGAN is robust to out-domain images with unseen textures, and can generate meaningful in-domain counterparts from these out-domain inputs. (a,b) Out-domain images as identity input. (c,d) Out-domain images as posture input.

GAN and the GAN-inversion encoder guarantees that the encoded latent vectors always lay in the latent space. Therefore, even for images with unseen visual attributes, $E$ still projects the image into the closest vector inside the latent space. Secondly, the VQ module will always reduce the shape feature-map into a certain number of learned object embeddings. It means the produced object-mask is always understandable to $G$, even for input images with unseen objects.

With the factors mentioned above, we show the synthesized images of PIVQGAN with unseen image domains in Fig. 6. We train our model on the Anime-Girl dataset in (a) and (c), which only contains animated female portraits. When we input a photo-realistic face as the identity image, the model automatically generates an "anime-girl" version of the input image. We can see that since the model has formed a solid prior bias on animated young girls, it translates all the faces into a young girl regardless of the input image's gender and age. On the other hand, it translates the other facial attributes correctly, including hair color, skin tone, and posture, because all these attributes are already learned from the training dataset. When we input real human faces as the posture reference, the model transfers the pose without any problem. Similarly, in (b) and (d), the model is trained only on photo-realistic animal images while we fed toy animal pictures as pose or identity inputs. Again, the toy's identity is well translated into a counterpart of the natural animal face, and our model captures the posture perfectly. In contrast, all the baseline models suffer from generating good quality results given the same training and input data. The out-domain performance reveals a promising performance of PIVQGAN on a zero-shot image domain translation task, which has not gained much research attention before to our knowledge.

## 5 CONCLUSION

We proposed the VQSN module and the self-supervised joint training scheme. The resulted model, PIVQGAN, tackles the problem of unsupervised posture-and-identity translation and improves the state-of-the-art performance on this task. The experimental results showed that our model outperforms prior baselines in both image quality and translation accuracy.

Beyond posture-and-identity translation, the proposed VQSN module also unveils the immense potential of fully unsupervised image segmentation and semantic understanding. Moreover, the robust performance of PIVQGAN on unseen-domain images gives hints to a new task of zero-shot image domain translation. We hope our model paves the way for future studies in this new direction.

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
