# OpenReview forum: "PIVQGAN: Posture and Identity Disentangled Image-to-Image Translation via Vector Quantization"
_ICLR.cc/2022/Conference — ICLR 2022 Submitted_

### Official Review · Reviewer_SbZB · 2021-11-01

**Correctness:** 3
**Technical Novelty And Significance:** 2
**Empirical Novelty And Significance:** 2
**Recommendation:** 5
**Confidence:** 4

**Main Review:**

**Strengths:**
- Experiments on 5 datasets
- Good qualitative results
**Weaknesses:**
The paper does not provide significant novelty on the generator architecture nor the training strategy. The proposed pipeline consists of using existing architecture (StyleGAN) and applying a series of feature operations without motivation. Below are my concerns about the work:
- My major concern regarding the paper is the proposed Vector-Quantized Spatial Normalization (VQSN) module which is an incremental improvement of feature aggregation without properly explaining the rationale behind such operations. It is not clear why such a module can learn the pose because there is no direct evidence that the model learns a disentangled representation.
- *Architecture choices:* It will be helpful if the authors justify the architectural choices. Specifically, justify the series of operations in the VQSN module over the “base feature map” that represents the pose. Additionally, the choice of taking 2 / 3 layers of a non-specified vector as a “pose-layer” is not motivated.
- In the self-supervised loss, if the only examples of the identity and poses come from transformed images (rotated, zoomed, …), how to make sure that the model is not overfitting on these examples? Or do you use a small subset of image pairs?
- Since the latent code combining the identity and the pose is obtained through the encoder E, then where is the inversion used?
- *Resolution:* The model produces images of smaller resolution (256x256), while other state-of-the-art approaches can produce images of higher resolution e.g. DATGAN [1] has resolution of 1024x1024 in the CelebA-HQ dataset and 512x512 in the AFHQ dataset.
- In the experiments, the variations VQ-1, VQ-2, and VQ-3 are not clearly explained and it’s hard to judge the benefit of such configurations

[1] Kwon et al. Diagonal Attention and Style-based GAN for Content-Style Disentanglement in Image Generation and Translation. ICCV 2021

**Minors:**
- In the paper the number 14 is used twice. First, as a hyper-parameter to represent the number of convolutions over the “base feature map” and as the latent vector combining the pose and identify features. Does the number 14 represent the same concept in both cases? Why specifically 14, does it cover all the variation of the face orientation and identities? Or, is it dataset specific?
- Can the authors clarify if the “Hosted embeddings” is a segmentation map? If so, this needs to be clearly specified in the paper
- Have you tested if the proposed VQSN can be used with other architectures e.g. PGGAN [2]
- Have you tried using other aggregations between $l^1_{[1:14]}$ and $l^2_{[1:14]}$ e.g. concatenation?
- “such as illustration synthesis and motion driving”… Please add references

[2] Karras et al. Progressive Growing of GANs for improved quality, stability, and variation. ICLR 2018

**Writing:** The presentation of the paper needs some revision. Overall, the text is not self contained and the concepts used are briefly described.
- Add reference when introducing concepts related to image-to-image translation e.g. has the pipeline consisting of an encoder E, generator G, and discriminator D been used before? What’s its advantage to just using G and D?

- Try to have a problem formulation explaining clearly what each of the components (E, G, and D) is expecting as inputs and how they are related along with corresponding dimensions

- The authors should specify if the method is specifically designed for StyleGAN or can be generalized to other models

- Try to separate the concepts from the implementation details

- Avoid using vague words such as: “certain number”, “we hope”, “ideally”. Be explicit and mention the values of each of your components.

- Use simpler words, please review and change some unclear sentences as highlighted below:
  - “we graft the latent sets” -> isn’t this a feature aggregation?
  - “Revised feature map”
  - “to be well communicated”

- Fig.2: Is a mixture between detailed architecture (top) and abstract concepts (bottom). Have them both separated and defined the variables used e.g. $l^p, f_b, f_{vq}$

**Summary Of The Paper:**

This paper presents PIVQGAN, a pipeline that tackles the image-to-image translation task. The contribution of the paper is two folds: (i) a module dubbed as Vector-Quantized Spatial Normalization (VQSN) that represents the pose and (ii) a self-supervised training scheme. Quantitative and qualitative results show good performances of the proposed pipeline.

**Summary Of The Review:**

Overall, I am leaning towards rejecting the paper. The current version of the paper needs to be improved, in particular: the novelty, writing, and the experiments.
The novelty of the proposed VQSN is incremental, however, I would consider changing the rating of the paper if the authors can implement the suggested improvements. In particular, improve the clarity of the paper and show evidence of the performance of their method against other state-of-the-art methods at resolutions 1024x1024 and 512x512.

---

> ### Author Response · Authors · 2021-11-10
> **Thanks for your review**
>
> Dear Reviewer
>
> Thanks for your review and your kind help to make our paper better!
>
> Q: "Motivation and intuition of VQSN layer":
> A: "Better control of the posture in the generated image" and "robustness to unseen out-domain image" are the two main motivations of our VQSN layer, and also the two advantages of our VQSN layer. Please consider the questions: which way do you prefer to represent the posture of the image: a vector of 512 dimensions or a semantic map telling the model which spatial position is what object? We find the latter works better and can be more intuitive, so it is our intuition to develop the VQSN layer.
>
> Moreover, please check Fig-6 in the paper and more results from the newly uploaded sup-material, the model disentangles the posture and identity attributes so well that it even can extract a pose of a human face (the model not even trained on such data) and apply the pose to an animal face with a totally different look than human (color, fur texture), we find it heartbreaking seeing the reviewer say "no direct evidence that the model learns a disentangled representation."
>
> Q: "Architecture":
> Please check experiment sec4.2 for more discussion of modeling choices. The posture layer choice is strictly following the inductive bias of convolution neural networks and experimental discoveries of prior work such as in StyleGAN. Given a CNN of 14 layers, the first 7 layers are easier to learn overall spatial
> attributes and the last 7 layers are easier to learn detailed texture-related attributes, so we select pose-layer from the first 7 layers. It is also discussed in our paper.
>
> Q: "Overfitting":
> A: we apply the transforms on all training images to create the training pairs, and the pairs are not fixed, new random transformations are applied on the same image at different training iterations, the model cannot remember or overfit on the "image pairs" because there are no two same "image pairs" for each image. During training, the image pairs are always new, creating on the fly.
>
> Q: "Encoding and inversion":
> A: Our encoder E will encode two images separately into two sets of vectors, each set has 14 vectors. Then we combine the vectors by picking vectors from the respective index in the two sets. The procedure is also presented inSec 3.1 in the paper.
>
> Q: "Resolution"
> A: Our model performs similarly well on higher resolutions: 512 and 1024. We have updated and uploaded code for training and testing for up to 1024 resolution. Higher resolution image results are also included in the sup-material. We will release pre-trained models at higher resolution later.
>
> Q: "14 layers"
> A: Yes, 14 always means the number of conv-layers and their latent vectors. It is more resolution-specific rather than dataset-specific. For a model trained on 256 resolution, we usually use 14 layers, if our model is for 1024 resolution, then we use 18 layers, two layers for one resolution. "Does it cover all variations?" It is a good question that can be applied to all GAN papers, no one answered it directly before to our limited knowledge, please, sincerely, don't reject our paper if we cannot answer it.
>
> Q: “Hosted embeddings”
> A: No, in the paper we use the word "segmentation-like representation" because it looks like segmentation but lacks precision, it is close to but not as proper as a "segmentation map". However,  it indeed learns the intuitive semantics well and can be used like segmentation to edit the generated image (please check the videos in sup-material).
>
> Q:  "VQSN for other architectures"
> A: Yes, the proposed training scheme and VQSN can be applied to any GAN structure, and we find it works well.
>
> Q: "Concatenation rather than grafting"
> A: No, we find concatenation is infeasible here, our model not only takes images as input to re-generate a new image, but it can also generate by itself, and it can generate without mixing from two resources. When generating without mixing from a single resource, the concatenation method makes it hard to design the model structure, because the input size changes.
> We do try interpolation between vectors, it works well and is presented in the paper.
>
> Q: "references for application"
> A: Will do, please also check the sup-material for a demo of our model on motion synthesis and semantic editing.
>
> Q: "Writing"
> A:  Thanks, we have revised the paper following the advice from the reviewer, and some missing information will be included in the appendix. Specifically, we add a missing reference to the training scheme of E,D,G; a detailed explanation of "graft the latent sets", which is not latent aggregation; and removed vague words like "ideally" and use more accurate sentences instead. We appreciate your help and effort, thanks again!

---

### Official Review · Reviewer_fSLH · 2021-11-02

**Correctness:** 4
**Technical Novelty And Significance:** 3
**Empirical Novelty And Significance:** 2
**Recommendation:** 6
**Confidence:** 5

**Main Review:**

Pros:
1.	The VQSN have is somewhat novel. The combination of vector quantization and spatially adaptive normalization is reasonable.
2.	The self-supervised training scheme is also interesting. By applying a series of augmentations to the real images and switching the latent code for original and augmented images, the model is forced to learning the smaller-scale pose information.
3.	The results look good.

The main concerns are in the experiment part which makes the reader quite confusing.

Cons:
1.	Why the qualitative results of ablation study and some baselines (such as MUNIT, TUNIT SNI DATGAN) are missed. It is importance to visualize the changes in generated images made by different components. Why quantitative results of CLUIT are missed?
2.	One important baseline is stylegan/stylegan2. Controlling the pose or identity in stylegan has been widely explored in recent works, such as GAN-inversion[3] stylegan-encoder[2], GAN latent disentanglement [1]. Comparing with stronger baseline may be better to show the advances of the proposed model.

[1]Shen Y, Zhou B. Closed-form factorization of latent semantics in gans[C]//Proceedings of the IEEE/CVF Conference on Computer Vision and Pattern Recognition. 2021: 1532-1540.
[2]Richardson E, Alaluf Y, Patashnik O, et al. Encoding in style: a stylegan encoder for image-to-image translation[C]//Proceedings of the IEEE/CVF Conference on Computer Vision and Pattern Recognition. 2021: 2287-2296.
[3]Pinkney J N M, Adler D. Resolution Dependent GAN Interpolation for Controllable Image Synthesis Between Domains[J]. arXiv preprint arXiv:2010.05334, 2020.
3.	Implementation details missed. Which generative architecture does the authors use for E, G, D? How does the VQSN module be plugged into the generator network?  It is very hard to re-produce results as the missed details.


**Summary Of The Paper:**

This paper proposed a Vector-Quantized Spatial Normalization (VQSN) module for pose-identity disentanglement in GAN’s training. The module combines vector quantization and spatially adaptive normalization. Then a self-supervised training scheme is introduced to increment the feature disentanglement. The experiments show the model is able to separate pose and identity and performs better than existing baselines.

**Summary Of The Review:**

Overall, the paper is novel in method while experiments are somewhat weak to support the claims.

---

> ### Author Response · Authors · 2021-11-18
> **Thanks for your review**
>
> Dear Reviewer
>
> Thanks for your review and your kind advice and help!
>
> We have added more qualitative results of the ablation study in the supplementary materials. For other baseline models such as MUNIT, they have a relatively big quality gap compared to the presented results. We do not include them in the main paper because we want to show more of our own results. CLUIT does not release the official code yet and we cannot reproduce their result (same as DATGAN at the time we submit the paper), we can not compute their scores and do more synthesis so we did not include them in the tables. But the CLUIT authors provide some images so we have included these images in our paper for quality comparison.
>
> The lack of disentanglement of the latent space in stylegan/stylegan2 is one of the main motivations of our work. Actually, Stylegan2's attribute variations are mainly entangled at the early layers, please check Fig.3 in the original StyleGAN paper, where they refer to as "coarse styles" control both the identity and the posture attributes (both face direction and human identity like skin color). The reviewer suggested methods (GAN inversion) working on disentangling the already entangled latent space after a generator is trained, while our method tries to learn a disentangled latent space for the generator from the beginning. We think the two types of methods are vertical to each other, GAN inversion techniques can directly be applied to our trained model. We have revised our paper to mention the papers suggested by the reviewer.
>
> We have updated our supplementary material, our code is included which is ready to run and test.

---

### Official Review · Reviewer_a2FQ · 2021-11-02

**Correctness:** 2
**Technical Novelty And Significance:** 3
**Empirical Novelty And Significance:** 3
**Recommendation:** 5
**Confidence:** 3

**Main Review:**

[strength]
The main idea of this paper is that automatically learn the arrangement mask, and use the vector quantization with spatially-adaptive normalization. This paper presents the generalized image translation ability on various datasets. Experiments show that the proposed disentangled framework is able to translate image style into different arrangements and styles. The detailed experiment verifies the disentangled-ability of the proposed framework. Paper writing and structuring well.

[weakness]
However, user studies should be conducted to verify the performance and conduct experiments on the high-resolutions.



**Summary Of The Paper:**

This paper focuses on the image translation problem, and proposes an arrangement and style disentangled framework that introduces the working logic by using vector quantization. It is a StyleGAN2-based method, with a vector-quantized spatial normalization. Besides, this paper also introduces a self-supervision method for the joint training between encoder and generator.  The experiments show effectiveness and interpretability. However, the experiments only use the 256p resolution, it is better to show more high-resolution results that compare with other methods.

**Summary Of The Review:**

This paper proposes a disentangled method for the image translation task, which is meaningful for the computer vision community. The method sounds works but the experiments are insufficient, more experiments should be conducted on high-resolution images.

---

> ### Author Response · Authors · 2021-11-09
> **Thanks for your review**
>
> Dear reviewer:
>
> Thanks for your review. Our model performs similarly well on higher resolutions: 512 and 1024. We have uploaded our sup-material with code and image results on higher resolution and will release pre-trained models for testing for up to 1024 resolution.
>
> The important reason we do not use 1024 to compute the metrics is: All the previous methods do not use 1024 resolution (as in DATGAN, StarGAN-v2, etc), because the pre-trained metrics model (such as inception net) is trained on 256x256 (actually less than 256), using higher resolution images to compute the metrics such as FID may lead to an inaccurate result.
>
> We also uploaded more images (trained at higher resolutions) and videos to showcase the ability of our model. Please check.

---

### Official Review · Reviewer_SuTz · 2021-11-02

**Correctness:** 3
**Technical Novelty And Significance:** 2
**Empirical Novelty And Significance:** 3
**Recommendation:** 5
**Confidence:** 4

**Main Review:**

Strengths
- The paper addresses an important task that enables many creative applications such as retargeting of videos without any explicit tracking
- The paper presents an unsupervised approach which makes it applicable to a much wider range of objects compared to supervised approaches which have to rely either on shape or identity annotations (although the latter is relatively easy to obtain from videos).
- The presented approach is relatively simple but produces very good results compared to baselines as demonstrated by both qualitative and quantitative comparisons.
- The model yields an interpretable shape representation in the form of a spatial mask that can be edited manually. This enables additional applications such as manual editing of these masks, which is also demonstrated in the experiments. It might also offer opportunities for  unsupervised segmentation of parts.

Weaknesses
- The related work discussion completely misses works on unsupervised keypoint learning. One of the main approaches in those works is to learn keypoints through image synthesis as a proxy task and many works such as [1-4] directly aim at disentangling shape (in the form of keypoints) from identity/appearance. They are even more relevant because this also provides an interpretable shape representation (keypoints) which can be used for other tasks such as manual shape manipulation and shape analysis, similar to what is presented in the work under review (and sometimes demonstrating additional capabilities such as swapping identity only on specific parts of objects, which would be a nice additional experiment so see how well this works with the presented approach). Additionally, there have also been works [5-7] which directly aim at disentangling shape and identity with a mask/part/segmentation-based shape representation similar to the one obtained in this work. Those works should be discussed and some of them also seem to be highly relevant for comparisons.
- Since the work is about disentangling shape, it would be desirable to see results on data which contain more shape variations. The main evaluation is limited to centered faces (of humans and animals), where the biggest shape/pose variations are limited to gaze-direction. While Fig. 1 also shows results on cars and churches, there is no further evaluation on that data and the qualitative results already suggest that the model is struggling much more on the car dataset. What I would really like to see however, is the performance on articulated objects, especially full body human datasets such as DeepFashion which have also been considered in previous keypoint/mask based approaches for shape and identity disentanglement as mentioned above.
- The experimental section is sometimes confusing and overall does not provide alot of insights. The text mentions that StarGAN-v2 and CLUIT have the best performance but CLUIT is not in Tab. 1 and both DATGAN and SNI are reported to have better FID scores than StarGAN-v2. Why does Tab. 1 report the scores for the VQ-1 variant on AFHQ but on CelebA it reports the number for the VQ-2 variant (Tab. 1 <-> Tab. 2)? Since the paper reports failure of Gumbel-Softmax, what about the argmax+straight-through estimator? The ablation study references Fig. 4 and 5 with respect to a failure of the VQ-2 variant but I fail to see how those figures connect to the ablation study. In fact, it is never mentioned with which variant those figures are produced (and as mentioned before, Tab. 1 reports number of different variants) and there is no qualitative comparison between the ablated versions. In general, there are too little qualitative comparisons and there should be a supplementary with qualitative results to be able to better judge the performance of the model and the different variants. In fact, the numbers reported in Tab. 2 obtained without using quantization (w/o VQ) outperform the baselines, so from this I really cannot judge whether the improvements over the baselines come from the proposed VQSN layer or from simply using a better architecture (StyleGAN) and training scheme (pretraining as GAN, training encoder to reconstruct latent but also encoder-decoder to reconstruct images). Given that the VQSN layer is the main contribution, its effect is discussed too little.

Minor:
- The reference keys are ambiguous, e.g. which of the works does Lee et al. refer to? There should be years/letters/etc. something to make them unambiguous and maybe follow a more common format. Also, putting reference keys after each of the method names instead of putting first all method names and then all reference keys in a list (Baselines paragraph in Sec. 4) would make it much more easy to find the correct references.
- Why not use color augmentations for the pose image of the generated pose-identity pairs? That should simulate the actual test case closer, no?
- The mixback LPIPS is a useful metric but also immediately suggests to use it for training similar to cycle consistency losses. Has that been tried?

References:
- [1] Jakab, Tomas et al. “Unsupervised Learning of Object Landmarks through Conditional Image Generation.” NeurIPS (2018).
- [2] Zhang, Y. et al. “Unsupervised Discovery of Object Landmarks as Structural Representations.” 2018 IEEE/CVF Conference on Computer Vision and Pattern Recognition (2018): 2694-2703.
- [3] Lorenz, Dominik et al. “Unsupervised Part-Based Disentangling of Object Shape and Appearance.” 2019 IEEE/CVF Conference on Computer Vision and Pattern Recognition (CVPR) (2019): 10947-10956.
- [4] Jakab, Tomas et al. “Self-Supervised Learning of Interpretable Keypoints From Unlabelled Videos.” 2020 IEEE/CVF Conference on Computer Vision and Pattern Recognition (CVPR) (2020): 8784-8794.
- [5] Hung, Wei-Chih et al. “SCOPS: Self-Supervised Co-Part Segmentation.” 2019 IEEE/CVF Conference on Computer Vision and Pattern Recognition (CVPR) (2019): 869-878.
- [6] Braun, Sandro et al. “Unsupervised Part Discovery by Unsupervised Disentanglement.” Pattern Recognition 12544 (2020): 345 - 359.
- [7] Liu, Shilong et al. “Unsupervised Part Segmentation through Disentangling Appearance and Shape.” CVPR (2021).

**Summary Of The Paper:**

The paper presents a generative model for unsupervised disentangling of shape and identity of images. The goal is to be able to combine two images of different identities such that the output depicts the person/animal/object of the first image in the shape/pose of the other image.

Overall, this is achieved with a encoder-decoder like architecture, where the latent code is divided into a shape and a identity part. The desired mixed image is then obtained by "grafting" the latent codes of the two images, i.e. taking the latent identity code from the first image and the latent shape code from the second image.

To achieve the disentangling of the two codes during training, the paper relies on two main techniques: Self-supervised generation of shape-identity image-pairs, obtained by shape augmentations of a single image. A vector-quantized-normalization (VQSN) layer which introduces a bottleneck on the pose latent code with the hope that it won't have enough capacity to capture identity information and therefore distill the pose information from an image.

Experiments compare the proposed model to previous approaches and ablated versions of the model and show applications enabled by features of the model such as smooth shape interpolation, shape editing via masks and robustness on out-of-domain images.

**Summary Of The Review:**

The paper presents a simple approach for an important topic and demonstrates both gains over previous works as well as interesting applications that are possible with the model. However, I think it misses an important line of research on unsupervised disentangling of shape and identity based on interpretable shape representations such as keypoints and masks (e.g. [1-7]). Given these prior works, the self-supervised generation of pose-identity pairs is no new contribution. The significance of the remaining contribution with the VQSN layer is difficult to judge, since the ablations which do not use this contribution also seem to quantitatively outperform existing approaches and also do not seem very much worse than the full model. Since there are no qualitative comparisons between ablated version, its value remains unclear. So while the presented approach produces very good results, these issues make it impossible to judge where that improvement really comes from and thus it is not clear if the claims regarding the contributions and their significance is correct or not. Therefore I currently lean towards rejection.

---

> ### Author Response · Authors · 2021-11-09
> **Thanks for your review**
>
> Dear Reviewer
>
> Thanks for your review and your kind help to make our paper better! According to your "Strength" section and "Minor" section, we can tell that you get most of the core ideas in this paper quite well, we really appreciate it!
> Blew are our replies to the weakness section:
>
> 1. Missing references:
> We will definitely add the suggested missing references. One excuse we would like to make here: Our work focused on Image generation, which differs from works [1-7]. We leveraged "unsupervised segmentation" to improve the image quality, synthesize robustness on the out-domain image, and better synthesis controalbilty, while we do not aim at improving the "unsupervised segmentation" task itself. However, we do believe if we could integrate the ideas from [1-7], it will make a better model for the image synthesis task. It is a promising future direction.
>
> 2. Datasets:
>
> 2.1. Unlike other works (StarGAN-v2, DATGAN, CLUIT) train human faces and animal faces with registered images, we also train our model without registering the images (human, animal faces has varied positions in the image with a larger background), while other models show clear image quality downgrade, our model is robust with almost no quality drop (an FID increase less than 1).
>
> 2.2. We do tested on datasets with more varied contents, like places2, unregistered cars, and churches, and the human body (iperdance dataset). We add the results in the newly uploaded supplement material.  Our VQSN layer is able to capture semantics such as car's wheel, front grill, front window; church's dome, main body, sky; and human body parts like arm and leg, etc. All this will be added to the appendix which we still working on.
>
> 2.3. Our model performs similarly well on higher resolutions: 512 and 1024. We have uploaded our code with training and testing for up to 1024 resolution and image results in sup-material, and will release pre-trained models.
>
> 3. Confusions:
>
> 3.1.CLUIT does not release the official code yet and we cannot reproduce their result, we can not compute their scores so we did not include them in the table. But the CLUIT authors provide images so we can include the images in our paper for quality comparison.
>
> 3.2. We have a dedicated discussion in Sec 4.2 on the presented structural designs shown in Table 2, and other reviewers find the discussion clear and helpful. We think the scores in Tab 2 make sense given their different structural design.
>
> 3.2.1. Importantly, "baseline" like DATGAN also uses stylegan as the backbone, so there is no other advantage of our model other than the proposed VQ layer and the training scheme. More importantly, the proposed training scheme (self-supervision with joint training) is also the main contribution of our paper, and it lead to the VQSN layer being able to learn semantics. Without this training scheme, the VQSN layer itself cannot properly learn well, as we show in Tab 2.
>
> 3.2.2. The training schema and the VQSN layer help each other to perform better, the VQSN layer can best take advantage of the training schema thus learning intuitive segmentations to help generate images and become more robust to unseen images.
>
> 4. Minors:
> Thanks! We have revised the references following your suggestions. Color augmentation is a good point, it defiantly worth trying, we will add it as an option in our code to be published. The mix-back can be integrated into the training, we will also include it into our code which will be released later.

---

### Official Review · Reviewer_dkA5 · 2021-11-03

**Correctness:** 3
**Technical Novelty And Significance:** 2
**Empirical Novelty And Significance:** 2
**Recommendation:** 6
**Confidence:** 3

**Main Review:**

Pros:
1. This paper is overall well written with sufficient details. It feels like not too hard to reproduce while reading the paper. The main idea is clearly explained with proper illustrations.
2. The main idea of combining vector quantization and spatially-varying normalization is intuitive and its effectiveness seems to be well supported by the experimental results when compared with related works and the visualizations. The learned masks do carry some semantic meanings as visualized.
3. I found the Section 4.2 quite useful for understanding the model. It seems like the author explored a number of alternative solutions including Gumbel-softmax and network variations to land on the version being reported.
4. Three applications nicely utilized the properties of the trained network with convincing results.

Cons:
1. The image contents used in the experiments seem somewhat limited, e.g. aligned faces or animal heads. And the resolution is also not very high, so I would not say I'm very impressed by the results. It may be worth considering more challenging settings, such as 1024x1024, articulated objects (humans and animals) and large-scale datasets like places2.
2. The proposed module is a nice combination of existing algorithms. While it seems to work well, but I would not be able to give a very high score for its originality.
3. The learned (unsupervised) masks are very interesting but their quality is not good enough to be useful as intuitive control signals for interactive image editing.
4. Typos: "pose-layers encode only the post information" --> "pose-layers encode only the pose information"
"We hypothesis this depends" --> "We hypothesize this depends"

**Summary Of The Paper:**

This paper presents a neural network module for achieving posture (shape) and identity (appearance) disentangling for image synthesis. The proposed module combines the vector quantization layer with the spatially-varying normalization layer so that the learned mask represents the shape while the "object" or the embedding for each mask label represents the shape-invariant features. Based on this module, this paper designs styleGAN-like generator with its discriminator and also an encoder that predicts the latent code from an input image. The training process is carefully managed with dedicated data augmentation methods to achieve the disentanglement. Results are reported on several commonly-used datasets with informative ablation study.

**Summary Of The Review:**

A solid work with nice results and insightful analysis. It has limited originality but can benefit from more challenging experiments.

---

> ### Author Response · Authors · 2021-11-09
> **Thanks for your review**
>
> Dear Reviewer
>
> Thanks for your review and your kind help to make our paper better! We can feel that you get the core idea of this paper quite well, therefore, to save your time, we will provide a brief reply only on the cons you listed.
> 1. The experiment datasets:
>         1.1. unlike other works (StarGAN-v2, DATGAN, CLUIT) train human faces and animal faces with registered images, we also train our model without registering the images (human, animal), while other models show clear image quality downgrade, our model is robust with almost no quality drop (an FID increase less than 1).
>         1.2. We do test on datasets with more varied contents, like places2 as you recommended, unregistered cars, and churches. We add the result in the newly uploaded supplement material.
>         1.3 Our model performs similarly well on higher resolutions: 512 and 1024. We have submitted our code with training and testing for up to 1024 resolution in supplementary material, and we will release code with pre-trained models for up to 1024 resolution.
> 2. When given the originality score, please also consider how we "combine" the "existing algorithms", what task we apply the "combination" on, and what we achieved by "combining" them. VQSN + cut-out self-supervision makes it possible to automatically learn semantics-like representations for better pose-identify disentanglement and higher image quality.
> 3. Agreed, but please check our newly uploaded video demo, we use the learned segmentation mask to control the generated image, It is a sincere video, apart from the goods, we also show limitations: 1. different VQ layers may control the same attribute of the generated image, 2. some VQ indexes may have unclear relation of generated image attributes. However, apart from the limitations, we believe our model marks a promising direction on image generation: semantic segmentation can be learned automatically without supervision, and it helps improve image quality and controllability.  We think our method provides a new and important direction for image editing without any supervision: how can we learn better semantics without labeled data such as semantic-segmentation masks?
> 4. Thanks, we have revised the paper.

---

### Decision · Program_Chairs · 2022-01-20

**Decision:**

Reject

**Comment:**

This paper offers a disentangled pose and identity representation for image to image translation.  The reviewers are borderline, but the AC finds the discussion by reviewer SuTz compelling, and agrees that the authors missed key references in the submitted manuscript.  In their rebuttal, the authors acknowledged the references were relevant, but believed their paper is not in the same area.  Overall this paper is borderline, but just below the threshold for acceptance in the opinion of this AC.